# Intelligent Safety Ergonomics: A Cleaner Research Direction for Ergonomics in the Era of Big Data

**DOI:** 10.3390/ijerph20010423

**Published:** 2022-12-27

**Authors:** Longjun Dong, Jiachuang Wang

**Affiliations:** School of Resources and Safety Engineering, Central South University, Changsha 410083, China

**Keywords:** safety system, intelligent safety ergonomics, safety management, big data era, system safety

## Abstract

Safety ergonomics is an important branch of safety science and environmental engineering. As humans enter the era of big data, the development of information technology has brought new opportunities and challenges to the innovation, transformation, and upgrading of safety ergonomics, as the traditional safety ergonomics theory has gradually failed to adapt to the need for safe and clean production. Intelligent safety ergonomics (ISE) is regarded as a new direction for the development of safety ergonomics in the era of big data. Unfortunately, since ISE is an emerging concept, there is no research to clarify its basic problems, which leads to a lack of theoretical guidance for the research and practice of ISE. In order to solve the shortcomings of traditional safety ergonomics theories and methods, first of all, this paper answers the basic questions of ISE, including the basic concepts, characteristics, attributes, contents, and research objects. Then, practical application functions of ISE are systematically clarified. Finally, following the life cycle of the design, implementation, operation, and maintenance of the system, it ends with a discussion of the challenges and application prospects of ISE. The conclusion shows that ISE is a cleaner research direction for ergonomics in the era of big data, that it can deepen the understanding of humans, machines, and environment systems, and it can provide a new method for further research on safety and cleaner production. Overall, this paper not only helps safety researchers and practitioners to correctly understand the concept of intelligent safety ergonomics, but it will certainly inject energy and vitality into the development of safety ergonomics and cleaner production.

## 1. Introduction

Ergonomics (or human factors) is the scientific discipline concerned with the understanding of interactions among humans and other elements of a system, which is an important part of the ILO’s occupational safety and health activities [1]. Safety ergonomics, as an important research direction of safety science and ergonomics, mainly studies the relationship among humans, machines, and the environment from the perspective of safety and uses system analysis methods to achieve optimal matching among them. In recent years, with the development of modern communication technologies such as 5th generation (5G) technology, cloud computing, and the Internet of things (IoT), big data has become a focus of attention in academia, industry, and government departments around the world. Big data has prompted many experts and scholars to reexamine research methods, guided thinking, and the technological innovation of scientific research [2]. At present, a new generation of information technologies represented by big data is being increasingly adopted in various fields [3,4,5], such as in safety science [6,7] and cleaner production [8,9]. In this context, a new era of safety ergonomics represented by intelligent tools is rapidly approaching, and safety ergonomics is facing a new round of revolution. 

The history of safety ergonomics has experienced a long development process. According to the development of system efficiency and relationships between humans and machines, the development of safety ergonomics can be summarized into three development stages: primitive ergonomics, ergonomics engineering, and modern safety ergonomics [10]. Since the beginning of human society, a simple interactive relationship has been formed between humans and machines [11]. In prehistoric times, to meet the survival conditions, humans enhanced their survivability by manufacturing tools. The initial and most primitive relationship between humans and machines was formed. After the industrial revolution at the end of the 18th century, labor-intensive and industrial-aggregating operations were advancing at a high speed, traditional manual manufacturing could no longer meet the development needs for productivity, and the requirements for tool reform became increasingly urgent. With the rapid advancement of mechanical physics research, early mechanized equipment was widely used in production and life. The development of electric power (the second industrial revolution) has greatly increased the number and types of production tools and equipment. To a certain extent, it also reduced the labor load of the personnel in the production system and increased the production efficiency of machines [12]. The First World War objectively promoted the development of ergonomics, and with the outbreak of the Second World War, the relationship between humans and machines gradually changed from "humans adapting to machines" to "machines adapting to humans," which also formed the theory of modern ergonomics engineering [13]. Since the 1950s, ergonomics has developed rapidly worldwide. However, safety issues in production have gradually emerged with the improvement of work efficiency [14]. In 1960, the concept of man-machine symbiosis was first proposed [15]. In the 1980s, the design of the interface and the method of interactive operations increasingly became a mainstream research direction [10,16,17]. In the 1990s, computer technology gradually matured, and computers slowly became a tool of control for manipulating complex production systems, and the process of mechatronics was accelerated [18]. Computer technology has changed the interaction forms and effects between traditional humans and machines, and it has accelerated the cognitive process of intelligent humans [19,20,21]. The human–machine relationship is also undergoing a new round of transformation, which provides a rare historical opportunity for the transformation and upgrading of safety ergonomics. The development history of ergonomics is shown in Figure 1.

With the advent of the Industry 4.0 era [22,23], the Internet of things and artificial intelligence (AI) have greatly changed the methods of working, producing [24,25], and thinking for humans. The complexity of the production system, as well as the interaction and collaboration between humans and machines is increasing [26,27], and the forms of human–machine collaboration to complete tasks has changed [28]; thus, the traditional accident causal model cannot satisfy the investigation and analysis requirements of complex system accidents and pollution [8], and the application of cleaner production faces huge challenges [29]. Therefore, the research content of safety ergonomics is no longer applicable to the analysis of new relationships among humans, machines, and the environment, and it is embodied in the following aspects: first of all, as intelligent communication technology overcomes the differences in time and space, the real-time operation of equipment and information integration becomes possible [30], and the traditional methods of interaction ways between humans and machines have been transformed into intelligent interaction. Secondly, the development and application of other disciplines [31,32] have injected more vitality into the research of safety ergonomics. At the same time, with the advent of the era of big data, information systems and various electronic computing devices are widely used in the practice of safety ergonomics. Whether it is the discipline of safety ergonomics itself or for the "humans, machines, and environment" elements included in safety ergonomics, big data technology has expanded its content and technology. Finally, compared with traditional classical disciplines, the disciplinary concept of safety ergonomics is lagging, and its theoretical foundation is still weak [33]. In particular, there are few cognitive analyses on the causes of systematic accidents, theoretical research and environmental pollution accidents.

Environmental research and public health are crucial for disaster control, environmental protection and human safety in production systems. The concept of environmental research and public health is being more and more widely used in different industries. However, there are still many problems in considering the safety and cleanliness of production systems using traditional safety ergonomics theories [34]. Therefore, based on the above considerations, in order to realize the coordination and unification among humans, machines and the environment in systems, and to promote the safe, correct, healthy, and efficient development of system operations, it is not only necessary to make full use of intelligent tools, but also to innovate the interactive modes among elements of systems. This is an inevitable trend to realize a new paradigm of safety ergonomics with intelligence as leading and core. In order to promote the development and practical research of safety ergonomics, we urgently need to answer the basic questions of safety ergonomics in the context of big data in a more detailed and systematic way. Therefore, in the second section, this paper proposes the basic concept of intelligent safety ergonomic (ISE), and analyzes the characteristics, attributes, contents and research objects of ISE through theoretical analysis. The third section puts forward the practical application functions of ISE. Finally, the future challenges and unsolved problems of ISE are analyzed in the final section.

## 2. Basic Questions of Intelligent Safety Ergonomics

### 2.1. What Is It?

Safety ergonomics is a branch of safety science environmental engineering, and its scientific research and teaching tasks have been widely carried out worldwide [35,36]. However, in view of previous studies, safety ergonomics still does not have a universally recognized definition [37]. The authors proposed a new definition of safety ergonomics [38]: safety ergonomics is the use of knowledge in physiology, psychology, environmental science, artificial intelligence, and other disciplines, with the goal of safety and comfort, taking ergonomics as a condition, making humans, machines, and the environment coordinate with each other to meet people's growing needs for a better life and working environment, so as to achieve safety conditions. The research object of safety ergonomics is shown in Figure 2.

With the continuous innovation and development of intelligent technology, the interconnected relationship between man—machine—environment is becoming increasingly complicated. In particular, the emergence of technologies such as intelligent human–computer interaction [39,40], biometric recognition [41,42], machine sensing, advance warning of disasters, and environmental adaptation [43,44,45] have greatly promoted safety ergonomics. The traditional definition is no longer applicable to the disciplinary system of safety ergonomics at this stage. Given the previous discussion, this section gives the following definition of ISE: intelligent safety ergonomics usually uses new-generation information technologies such as cloud computing, big data, 5G, and the Internet of things (IoT) as tools in combination with the knowledge of human factors, informatics, cybernetics, bionics, and environmental sciences to meet people’s growing demands for a better life and work as a starting point. It then focuses on the thorough integration of safety and environmental concepts into the design, implementation, operation, and maintenance of the human–machine–environment system throughout the life cycle, to achieve the intrinsic safety and cleaner production of the system as the goal, as to realize the self-adaptation, self-training, self-maintenance, self-learning, and self-optimization of the "human–machine–environment–information" system. The conceptual diagram of ISE is shown in Figure 3.

The relevant theories in ISE are not unfamiliar. The new generation of information technology can be traced back to the field of computer engineering. The foundation of related disciplines can also be found in traditional human factors engineering. Some concepts of intelligent disciplines have also been proposed previously, i.e., intelligent mining [46,47], intelligent ergonomics [48], intelligent human factors engineering [49,50], etc. However, the concept of ISE is proposed for the first time. ISE considers the full life cycle of the system, which is completely different from the methods of traditional safety ergonomics.

With the development of information technology, especially in the context of big data, audio, video, and other massive data "blowout" growth [51], a wide range of data collection covers various fields of safety ergonomics research and human–machine–environment system data perception, and the processing technology is more intelligent; the internal data structure is also broader and more diverse. The differences between traditional safety ergonomics and ISE can be summarized in Figure 4.

### 2.2. What Are Its Characteristics?

Intelligence is the general term for intelligence and ability. At present, "intelligence" has been widely used in many fields [52,53], and it can be understood as a convenient tool to meet various needs in production and life, with the help of technologies such as computer networks and the IoT. Currently, breakthroughs have been made in the fields of machine vision, speech recognition technology, and equipment motion control, which are gradually integrated into other industries [25]. The development of intelligent industry and technology has promoted the diversification of human–computer interaction methods, and it has also improved the efficiency of these interactions. To a certain extent, it has also stimulated the development of safety ergonomics.

According to the basic concept of ISE, it can be considered that, ISE mainly coordinates the relationship among the elements of the system, integrating safety and the environmental concept into the whole life cycle (design, implementation, operation, and maintenance) of the system to the maximum extent. This is done to solve the problems of low safety performance, serious pollution problems, low degrees of intelligence, low self-adjustment and self-adaptive ability, the lack of guaranteed of intrinsic safety, and the difficulty in realizing safety control during the operation for system, thereby enriching the theoretical system of safety science. On a macro level, ISE has the following five main characteristics (advantages): 

(1) The main research subject of ISE is still the "human–machine–environment" system. Realizing the efficient integration and optimal matching among the elements in a system is always the research focus in the field of ergonomics. However, with the development of information technology, the form of system interaction is gradually changing, and the previous normative interaction is also changing toward intelligent interaction. Especially with the development of sensor technology, "sensing" emerges in each interactive interface, and the information generation mechanism, transmission method, and interactive information performance characteristics among the elements of a system have undergone significant changes. Multi-channel information fusion interaction has gradually become a new form of interaction.

(2) The ultimate goal of ISE is to achieve intrinsic safety and cleaner production. It is difficult to achieve these goals in the construction of traditional safety ergonomics, but with the rapid development of information technology, intelligence has given us new technical methods. The goal of ISE is to carry out safety control and environmental protection measures, integrating safety and environmental concepts into the system throughout the life cycle of the system, so as to ensure the reliable operation of the system before the addressing critical safety and the clean state, thus achieving intrinsic safety and cleaner production. For example, in the life cycle of mine production, the degree of risk surrounding it can be judged by monitoring the subtle changes in rocks [54], employing the early implementation of safety control measures. It can be seen that with the use of the tools and theories of intelligent safety ergonomics, we can complete the safety measures that cannot be guaranteed by traditional technical means, and thus achieve intrinsic safety.

(3) The tool of ISE includes a broad new generation of information technology. In view of the existing intelligent manufacturing [55,56], intelligent visualization, and other means, the research on the integration of the new generation of information technology and safety ergonomics is still stuck in the theoretical integration of interdisciplinary theories, its in-depth integration and application need to be further developed.

(4) ISE is a discipline that combines theory and practice to guide the system to carry out safety and environmental control measures. First of all, through the theoretical analysis of ISE, it can guide management personnel to carry out safety practices, to further sublimate the essence of ISE. Secondly, ISE guides safety managers to carry out safety control practices, determine the safety state by analyzing the energy interaction among humans, machines, and environment, and implement safety and the environmental restraint measures through safety and environmental state feedback, thereby maintaining the dynamic balance of the system.

(5) ISE includes safety science, environmental science, and data science. First, the multi-element interactive fusion form widely existing in ISE is essentially the dissemination of digital information flow. Secondly, relying on the theory of ISE to solve practical problems, it is necessary to ensure that humans and devices have the capabilities of data collection, data analysis, and data processing, and then use data-driven approaches to develop related ISE content.

### 2.3. What Are Its Attributes?

Obviously, ISE is an applied branch of safety science and environmental engineering. Like other disciplines of safety science, ISE is the product of traditional safety ergonomics and intelligence. In addition, ISE intends to study the interaction and fusion method among the internal elements of a system, obtain the risk information of the system through intelligent perception, and then provide safety managers with a starting point and foothold for safety and environmental control methods, to achieve intrinsic safety and cleaner production. The specific attributes of ISE can be summarized in the following aspects [38]:

(1) ISE is a discipline that studies the coordination mechanism of the "human–machine–environment–information" system under the background of safety and the environment. With the advent of the information age, the rapid development of informational products has given birth to big data technologies based on massive amounts of data. In this context, the coordination mechanism among the traditional "human–machine–environment" systems has undergone fundamental changes, the communication among the various subsystems has changed, and the pattern and relevance of the identification data require a new safety ergonomics methodology.

(2) ISE is primarily affiliated with safety science and environmental engineering. First of all, its basic theory is still based on safety science and environmental theories. At the same time, realizing the intrinsic safety of the system is the ultimate goal of ISE. In addition, the theory of safety science involves studying the law of movement of safe things [57], and ISE is the study of the law of movement of the safety system with intelligent tools. It can be seen that ISE is still subordinate to the general and specific relationship between safety science and environmental engineering.

(3) ISE is a multi-disciplinary integration of safety and environmental science. The birth of ISE is an inevitable trend in the current information society, and it also involves the infiltration of other multidisciplinary disciplines into the field of safety and environmental science.

### 2.4. What Are Its Contents?

According to the research content, the content of ISE includes the following aspects [38].

(1) The basic theoretical level of ISE belongs to the upstream research content. The focus is on the research methods, analysis methods, design methods, basic principles, and subject attributes of ISE. Among them, the methodology of ISE focuses on the research on the scope, principles, theories, and methods of ISE to solve practical problems. It is usually based on ISE as the main body, summarizing and refining the research methods and paradigm systems that can solve practical problems. The basic theory of ISE refers to the research content and principles of other interdisciplinary subjects in the process of discipline construction. The discipline system of ISE mainly involves the basic issues of discipline construction. The upstream research content of ISE can be regarded as the scientific research on the subject. The basic theoretical level of ISE is shown in Figure 5.

(2) ISE is an interdisciplinary subject, and the research on the basic level of its application is mainly conducted to extract scientific knowledge from the subjects it contains, and then to carry out research work on the basic problems of its field. The main research content of "intelligent humans," includes the decision–making skills, cognition, behavior, and physiology of humans, which can realize the functions of information perception [58,59], recognition, and prediction. The research purpose of "intelligent machines" includes intelligent perception, monitoring, detection, operation, and maintenance, etc. The research purpose of intelligent environmental is to realize the dynamic perception of the environment, and then to clarify the internal and external conditions of the production system. The specific content includes the static perception of the physical environment, the dynamic perception of the hidden danger environment, etc. [60]. The intelligent interaction can also be called the intelligent fusion technology of human–machine–environment elements, which refers to the situational cognitive function of intelligent humans and intelligent machines, and then to realize the semantic analysis of information parameters and intelligent processing of data information among each element. The application of the basic level of ISE is shown in Figure 6.

(3) The purpose of any new subject is to solve practical problems. As far as ISE is concerned, the purpose of its practical application is to identify critical risk system information by identifying the failure and fault state of the system and to carry out safety control activities through intelligent monitoring, intelligent detection, and intelligent prediction. The practical application level of ISE contains many fields; specifically, it includes functional matching design for humans and machines, the optimal design of an intelligent interface, optimal intelligent design for work efficiency, and optimal intelligent design for intrinsic safety, etc. By carrying out research work at the practical application level, the risk level of the human–machine–environment system is converged, and then measures are provided for achieving the intrinsic safety and cleaner production of system. Practical application level of ISE as shown in Figure 7.

### 2.5. What Are Its Research Object?

Any production system always contains three parts: the human body, the mechanical equipment or body, and the environment in which they are located. Different from traditional safety ergonomics, ISE adds a new element-information based on the original research elements. The research object of ISE is shown in Figure 8.

Humans in traditional safety ergonomics are defined as natural persons who possess autonomy and creativity and who can explore and manipulate various devices, as well as make autonomous decisions. With the continuous acceleration of the intelligentization process, the labor cost in the production system has been greatly reduced, and the efficiency of production has also been fully improved. Some tedious tasks that can lead to fatigue characteristics in the human body are gradually being taken over by intelligent equipment. ISE proposes that the era of intelligence creates higher requirements for humans, and maintains that they are still the controllers of these intelligent machine. In particular, intelligent technology makes the production system more complicated, and human judgment is more susceptible to error. Therefore, humans are no longer individuals who deal with a single job in the traditional human–machine matching relationship, but "intelligent people" who can carry out intelligent decision-making tasks and control and break through the obstacles of human–machine interaction.

In ISE, a “machine” can be defined from a macro perspective as a mechanical device or energy carrier that can match human functions in a production system and can satisfy some specific functions. From the perspective of safety, the machine here is not only a specific mechanical device but also a macroscopic mechanism that can produce an unexpected release of energy. With the development of big data and the Internet of things technologies, intelligent devices continue to be popularized. These devices can realize functions such as computer processing, intelligent judgment, intelligent collaboration, and multi-channel interaction through sensors of the Internet of things, mobile networks, and other technologies. In turn, it can actively complete various tasks assigned by the production system.

The environment in ISE often includes the natural environment in which the humans and machines are located, such as humidity, temperature, and toxicity, which can be understood as the characteristics of the external conditions in the process of the human operation of the machine. In addition, it also includes the safety culture, safety atmosphere in the production system, and safety spirit. From a micro level, it can be understood as all environments that hinder or promote the integration of humans and machines. The psychological environment that affects human thinking is also included in this category.

In the phase involving the integration of humans, machines, and the environment, the timely regulation of the elements is essential to ensure the stability of the system and to achieve safety control. The interaction process of these three elements often requires a transmission carrier to facilitate communication, and information serves as such a function. It can be considered that the internal energy transmission and action mechanism of any abstract system are often transmitted by information, but it is different from traditional safety ergonomics, as the information in ISE is an interactive information flow created by external intelligent technology. The generation mechanism, action mechanism, transmission method, and transmission direction of traditional information flow are completely different. In addition, another reason why ISE is different from traditional safety ergonomics is the efficiency of information flow among the elements. The propagation mechanism of information affects the integration strategy of the human–machine–environment.

## 3. Practical Application Functions of ISE

From the definition and characteristics of ISE, the basic tasks of ISE can be obtained as follows: under the premise of giving the safety and environmental concept the leading role, the current emerging information technology tools are used to optimize the information flow dimension among humans, machines, and the environment to improve the intelligent coordination function of the system and to integrate it into all aspects of system analysis to achieve intrinsic safety and cleaner production. From the perspective of ISE practice, it pays more attention to the efficiency of human-computer interaction, the accuracy of system information transmission, and the interactive form of functional matching between elements. The coordination method and specific interpretation are shown in Figure 9.

### 3.1. Practical Application Functions of ISE

In order to further realize the above functions and understand the necessity of ISE, it is necessary to conduct an in-depth analysis of its functions. The so-called practical application function of ISE mainly refers to its safety and environmental control effect on human production and life. According to the focus of ISE, its practical application functions can be summed up as safety and environmental information perception, safety and environmentally intelligent monitoring, safety and environmentally intelligent detection, safety and environmentally intelligent prediction, and safety and environmentally intelligent operation and maintenance.

(1) Safety and environmental information perception. This is a basic function for the development of ISE application practice. Information is an important medium for ensuring functional matching among these three elements. ISE involves sensing technology; while people understand external information through physiological functions, the elements can also use intelligent sensing technology to complete the characterization and reasoning of external information through computing attributes to realize the interactive fusion among elements. For example, in mine production, we can determine the degree of risk by identifying potential risk areas and ensuring safe production in these areas [61,62,63]. 

(2) Safety and environmentally intelligent monitoring. This function includes relying on intelligent sensor technology to capture human physiological characteristics, health status, behavioral activities, language activities, location information, fault information, internal defects of mechanical devices, the information accessibility effect, the human–machine matching effect, and external environments such as the production environment temperature, wind speed, etc. According to the feedback results, the adaptive function adjustment in the intelligent coordination mode is used to cause the system to reach the best functional state. Existing studies have shown that it is feasible to use the Internet of things and cloud computing to establish a production early warning system for real-time monitoring, which also provides a technical guarantee for the feasibility of ISE to guide engineering practice [64]. Moreover, in the process of mine production, by using the current advanced positioning method (such as complex structure positioning [65,66,67,68]) to quickly determine the location of rock failure, it is also possible to determine the potential risk location in time according to different environments [69,70,71], thus realizing the purpose of early warning [72,73].

(3) Safety and environmentally intelligent detection. The safety status and stability of the system operation have a certain degree of connectivity, and there are hidden dangers in the operation process at any given time. To ensure the operation of the system, ISE requires regular intelligent detection activities, so as to determine hidden risks that may evolve into disasters, as well as the timely implementation of safety control activities aimed at controlling the potential risks of the system.

(4) Safety and environmentally intelligent prediction. Due to time constraints, the occurrence of accidents and disasters often lags. The monitoring function is continuous and occurs in real–time, and it does not have include a predictive function. However, as ISE can realize system inspection, detection, and data recording, "intelligent humans" can complete the evolutionary reasoning of the system through personalized cognition and then realize intelligent prediction.

(5) Safety and environmentally intelligent operation and maintenance: ISE emphasizes that systems can rely on big data technology and machine learning, collecting massive data (including logs, business data, system data, etc.) from a variety of data sources for real-time or offline analysis, thus enhance the ability of traditional operation and maintenance through initiative, humanization, and dynamic visualization to realize automated and intelligent decision–making.

### 3.2. Research Steps of Intelligent Safety Ergonomics

Intelligent safety ergonomics emphasizes providing intelligent perception functions to the system. The intelligent perception of the system mainly uses intelligent devices to monitor the relevant parameters or indicators that affect the unsafe behavior of humans, the unsafe state of machines, and the unstable changes of the environment during the operation of the production system, and analyzing, processing, and comparing the obtained results, to make dangerous or non-dangerous judgments regarding the system. This is a brand-new discipline whose research process, research, and application steps need to be explored continuously. Here, the general steps of the application of ISE in the practice of system safety control are presented.

(1) Analyze the characteristics of the researched system object, and select the relevant theories in ISE.

(2) Based on the cross-scientific theories contained in ISE, initially ascertain the common safety laws in the safety control process of the system.

(3) Propose a theoretical model of intelligent and safe ergonomics. ISE theoretical model is a theoretical framework that guides safety control technology to capture the critical safety state existing in the human–machine environmental system.

(4) Obtain the data sequence of the variable parameters of the critical safety state according to the safety information perception function of the intelligent system.

(5) Monitor changes in the critical safety state parameters, establish predictive models and causal chain analysis modules, and reveal the dynamic change law of the critical safety states.

(6) Combined with the dynamic change law of the critical safety state, discover the evolution process of the failure state for the system, use data dimensionality reduction methods to realize the dynamic visualization of the catastrophic evolution process of the system, and realize intelligent safety prediction.

(7) Combined with the intelligent cognitive decision-making mechanism, obtain the spatiotemporal strong mapping sequence of the node causality in the catastrophic evolution process, solve the catastrophic evolution link in reverse, analyze the final safety state and hidden danger position of the human–machine environment system, and carry out a scientific safety control strategy.

## 4. Prospects and Challenges of Intelligent Safety Ergonomics

The information processing technology and tools derived from the big data era have greatly enriched the development of safety science and the connotations of ergonomics, and they also provide new development ideas and methodology for safety ergonomics. First of all, ISE emphasizes the study of safety and environmental issues in the whole life cycle of the human–machine–environment system from the perspective of safety. Secondly, the advent of the big data era has overturned the traditional analysis methods of the systems, which has further expanded the research scope of these systems. Most importantly, the development of many new-generation artificial intelligence technologies has given birth to a broad, multi-dimensional system. With the help of the coordinated control function of the intelligent system, it can further provide a basis for solving the system function problems or the management of decision–making, reducing the risk of system disasters and environmental pollution to continuously improve and enhance the coordination and safety cleanliness of the system.

Mathematician David Hilbert said: “The branch of mathematics must be able to continuously produce new problems before it can be considered vigorous. Research fields that cannot ask questions are equivalent to a gradual death [74]”. With the rapid development of high-precision technology today, any discipline will have unresolved problems in its development process, and these problems also promote the progress and development of the discipline. ISE is the development and continuation of traditional safety ergonomics, and it is also an inevitable trend in the development of disciplines in the era of big data; thus, the future development of ISE still faces the following challenges.

Challenge 1: Intelligent perception and sensing technology of the human–machine– environment system.

The intelligent perception of the human–machine–environment system is the basis and key to realizing safe decision-making and control. Few studies regard it as a whole and reproduce it in the digital world, and most of the current sensing technologies usually realize intelligent perception through port and software analysis. As one of the core technologies in the era of big data, sensing technology is the key to the research of analog-digital signal processing, edge computing, and big data platform architecture. Although sensing technology has been applied to many fields, it is still difficult to build an intelligent system with integrated functions of sense, knowledge, and connection. 

Challenge 2: Data acquisition and the information fusion mechanism of an intelligent system.

The advent of the era of big data has brought opportunities for the innovation and iteration of disciplines, as well as new tests for discipline practice. As big data technology guides a new round of disciplinary revolution, the ties between disciplines are moving closer and closer. However, it is a complicated, multi-dimensional, large-scale structure, which is difficult to clean, and which includes data and information island problems, heterogeneous multi-source multi-modal data fusion problems, and system mutation problems. At the same time, heterogeneous data and the fusion of multiple fields will have an impact on the safety control of the system. For example, in the process of mine production, by data collection and information fusion, the different risk levels of the production stage can be determined immediately, and the location of rock fracture and spatial evolution characteristics can be confirmed under complex stress conditions [75]. Therefore, the discussion of the data acquisition and information fusion mechanism of intelligent systems will become a major problem and challenge for ISE from cognitive theory to practical application.

Challenge 3: Design and development of a functional program of an intelligent and safe human–machine–environment system. 

At present, intelligence has gradually become popular in visualization and data mining technology, but the degree of intelligence and automation is still not optimistic, especially for the dynamic quantitative characterization of a system disaster evolution mechanism, which is still difficult to achieve. In addition, the existing perception analysis technology still exhibits has the problems of a lack of information timeliness and a long time-consuming process. The design and development of the functional program for the system is still the focus of the future development of the discipline. 

Challenge 4: High-speed storage and processing of intelligent system data.

Over the past 60 years, few scholars have focused on the use of human factors and ergonomic design to reduce system risks through effective interventions. Indeed, when the amount of data shows "blowout" growth, the efficiency of the useful information obtained from the data will be greatly reduced. Therefore, the high-speed storage and processing of data are also very important for the construction of the data architecture and the realization of the algorithm structure. It should be noted that due to the fact that big data has the characteristics of multiple sources, heterogeneity, and rapid change, the research on the computing paradigm of big data will also be the focus of future discussion because this is essential to the fine management of data quality and the collection of effective information.

Challenge 5: Implementation plan for the intrinsic safety and cleaner production of the full life cycle of the intelligent system. 

In traditional ergonomics, since machines cannot realize the functions of self-diagnosis and self-feedback regarding their health status, errors will also occur regarding the fatigue operations of personnel, in this case, it is difficult to guarantee safe and clean production. With the development of big data tools, all of this has become possible. In addition, the purpose of building ISE is to explore the risk status of the human–machine–environment system and to achieve intrinsic safety and clean production through control methods. For example, in the production process of mines, microseismic monitoring technology, image recognition technology, artificial intelligence, and intelligent sensors can be used to develop intelligent mine driverless cars, thus realizing the intelligence of mining operations. Not only can it protect the safety of workers and equipment, it can also provide technical support for the safe and efficient recovery of resources [25]. Therefore, a reasonable implementation plan for intrinsic safety and cleaner production is also a challenge that ISE needs to face.

Challenge 6: Privacy protection and refined management of data.

Data is the fifth major production factor after land, labor, capital, and technology. Data security is a sensitive area in the process of data mining and utilization. In the era of big data, the statistical process of ergonomic safety data is more complete, and the scale of data continues to expand. Ensuring the confidentiality, integrity, and availability of the production system is also the data security solution pursued by ISE. Therefore, in the process of obtaining and using massive amounts of production data information, the protection of data privacy is both a technical issue and a social issue. For example, in order to ensure the sustainability of the mine production system, it is possible to analyze the multi-source heterogeneous data of the mine by establishing a human–machine–environment system evaluation method to realize safety control [76,77,78,79]. 

Challenge 7: Interaction and awareness insight mechanism for a human–machine–environment–system.

The key to the integration of system elements is to learn the behaviors of each aspect intelligently, realize awareness insights by finding common laws among fused data, and then construct a decision-making intent space to implement safety control. However, due to the complexity of the fusion mechanism, the interference of conscious behavior, and the ambiguity of state recognition, it is difficult for elements to realize each other's intentions. Therefore, it is very important to research a consciousness insight mechanism to realize the intelligentization of the system. In addition, the intelligent interaction process is subject to the existence of a wide range of information, and people's perceptions may be affected. The process of macroscopic intelligent mechanical manipulation may inhibit the understanding of information. Therefore, the technology to eliminate the visual noise regarding interaction interface is also a major challenge for ISE.

Challenge 8: Timeliness of information feedback regarding intelligent and safe ergonomics.

The limitations of big data technology, the complexity of multi-channel feature fusion for information transmission, and the uncertainty of human cognitive decision-making may cause a certain delay in the safety decision-making control for a system. Therefore, to improve the intelligentization process and efficiency of safety ergonomics, it is necessary to use artificial intelligence technology to speed up the design and development for new algorithms to ensure the timeliness of information feedback. Especially with the emergence of new imaging hardware and artificial intelligence chips, the design and development of intelligent algorithms for different chips and data acquisition devices is also a challenge.

Challenge 9: Operation and maintenance of the intelligent equipment of the system and the real-time control of the remote network.

On the one hand, intelligent operation and maintenance is a key technology to improve production efficiency, reduce safety management costs, and ensure safe production quality. It is also an important technical method to use intelligent technology to discover and solve abnormal operations of the system. However, due to the widespread existence of big data technology, wireless networks, and Ethernet networks coexist, and barriers to interconnection and intercommunication have increased, intelligent control and operation, as well as the maintenance of equipment groups are relatively difficult. On the other hand, it is difficult for the prior method to perform remote networking control of multiple device clusters through one control terminal, which also brings severe challenges to the intelligence of the machines. Dynamic decision-making component of intelligent equipment group operation and maintenance of the system is shown in Figure 10.

Challenge 10: Intelligent interactive large-scale image perception and high-speed transformation technology. 

The intelligent interaction of humans and machines is the focus of the research content for ISE. In order to realize the visualization of all data, it is necessary to seek common visualization technology and to ensure that intelligent machinery is highly sensitive to massive amounts of data, then presenting a large amount of safe information in different text forms. Therefore, the development of the intelligent interactive large-scale image perception and high-speed transformation technology of the "human–machine–environment" system is also the key content and major challenge requiring discussion.

## 5. Conclusions and Limitations

With the development of the Internet of Things and artificial intelligence technology, the way of life and production for humans are realizing the transition from informatization to intelligence. Systems are more complex, and the forms of interaction for the constituent elements are more diverse. Traditional safety ergonomics is no longer suitable for analyzing the elements in the new production relationship, and the iterative update of the theory is imminent. Intelligent safety ergonomics, as a new branch of safety science and environmental engineering, aims to thoroughly integrate safety and environmental concepts into the entire life cycle of the design, implementation, operation and maintenance of the human–machine–environment system; this is of great significance for achieving intrinsic safety and cleaner production. Of course, the research in this manuscript is not enough. The discipline construction of ISE still has a long way to go. Through analysis and research, we sorted out the following conclusions:

(1) Intelligent safety ergonomics, as an emerging notion created by the intersection of intelligent technology and safety ergonomics, is an inevitable trend for the iterative renewal of disciplines under the background of the era of big data, and it is also a cleaner research direction for ergonomics in the era of big data.

(2) With the aid of analyzing the definition of ISE, this manuscript answered the basic questions of ISE. At the same time, for further interpreting ISE, the practical application functions of ISE were systematically elaborated, and the future challenges and unsolved problems were analyzed. 

(3) Although the concept of ISE has been proposed, its further research still faces many challenges. It must be made clear that this study is a preliminary exploratory study of ISE, and it does not provide comprehensive answers to its theoretical questions. It is still a lack of suitable case studies on how production systems can be used effectively. It can be seen from the challenges faced by ISE in the future that more extensive research on the scientific issues of the subject is needed, it is an important element that needs to be continued in future research to promote the development of theory and practice for safety science.

## Figures and Tables

**Figure 1 ijerph-20-00423-f001:**
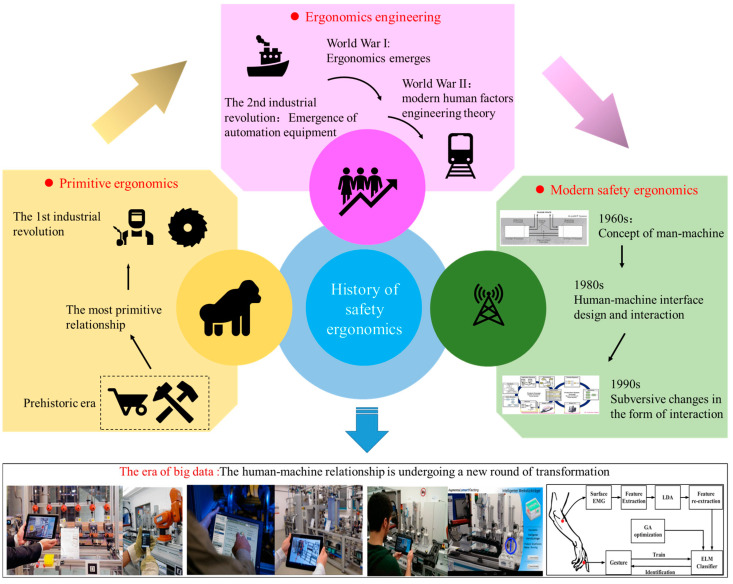
The developmental history of ergonomics. The development of safety ergonomics can be summarized into three development stages: primitive ergonomics, ergonomics engineering, and modern safety ergonomics.

**Figure 2 ijerph-20-00423-f002:**
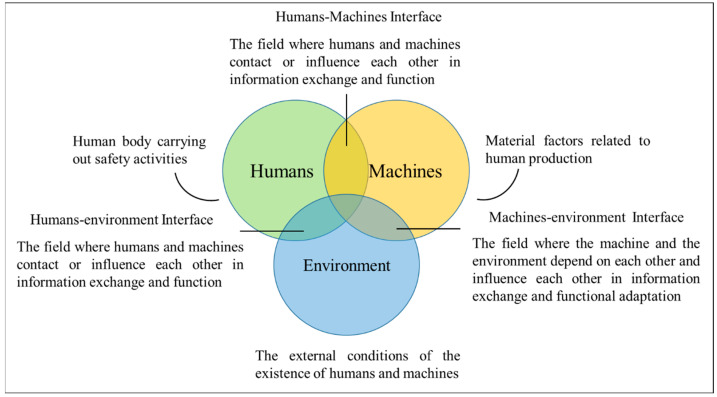
Research object of safety ergonomics. The main research objects of safety ergonomics include humans, machines, and the environment, as well as the interactive interface between various elements.

**Figure 3 ijerph-20-00423-f003:**
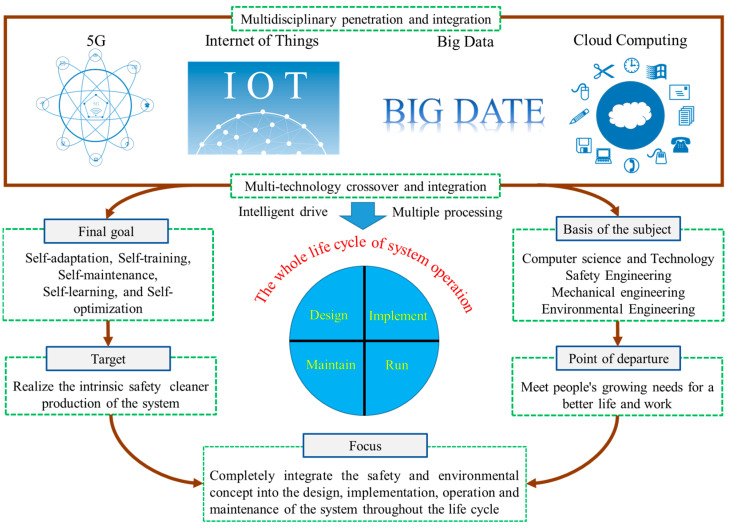
The realization tools, basis of the subject, and point of departure, focus, target, and final goal of ISE.

**Figure 4 ijerph-20-00423-f004:**
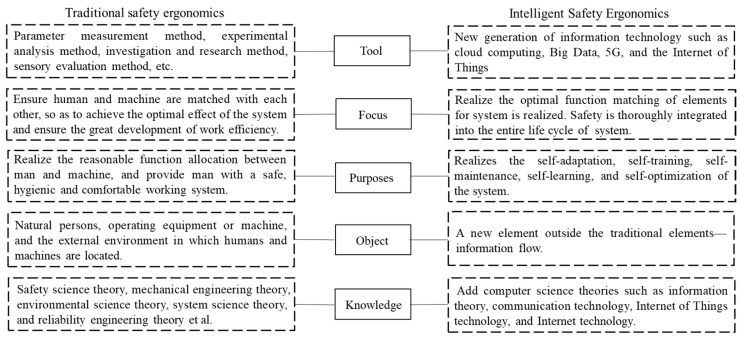
The difference between traditional safety ergonomics and ISE. The differences can be analyzed according to the tools, focus, purposes, objects, and knowledge.

**Figure 5 ijerph-20-00423-f005:**
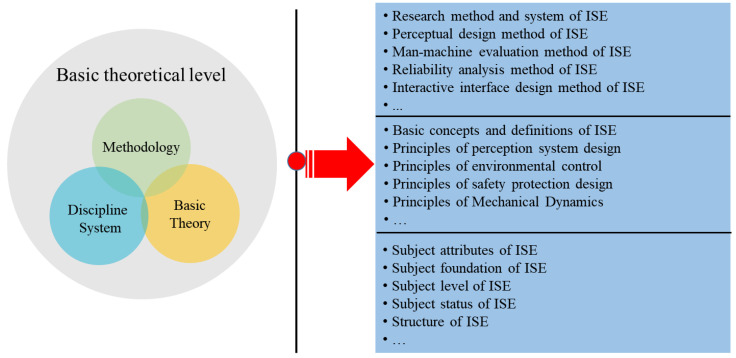
Basic theoretical level of ISE. Basic theory mainly includes three parts: methodology, basic theory, and subject system.

**Figure 6 ijerph-20-00423-f006:**
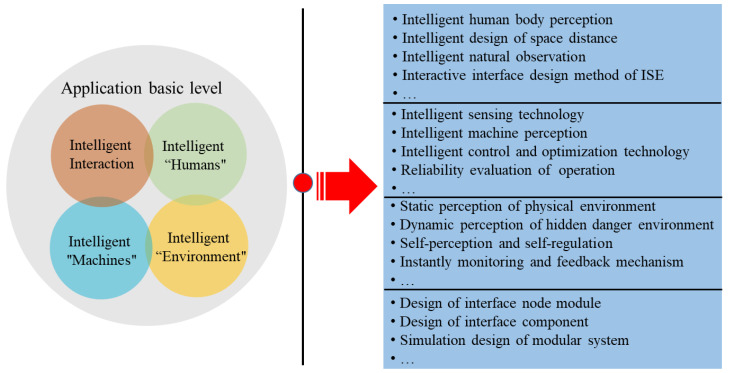
Basic application level of ISE. Application of the basic level mainly includes intelligent interaction, intelligent humans, intelligent machines, and the intelligent environment.

**Figure 7 ijerph-20-00423-f007:**
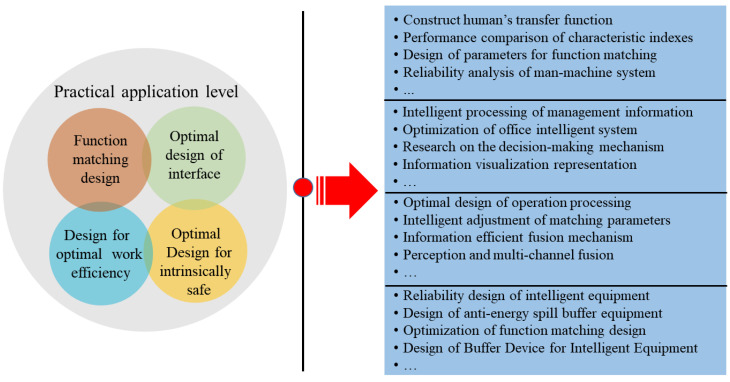
Practical application level of ISE. Practical application level mainly includes function matching design, optimal design of interface, design for optimal work efficiency, and optimal design for intrinsically safe.

**Figure 8 ijerph-20-00423-f008:**
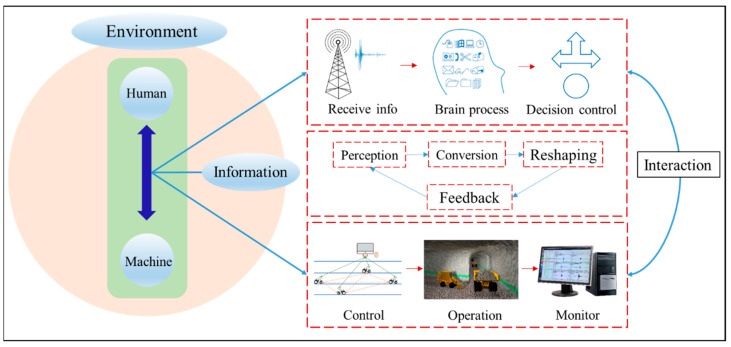
Research object of ISE. After receiving the information, humans use their brains to process original information to make safety decisions. Through interaction between humans and machines, intelligent control of the machines is realized, and results will be presented on the display.

**Figure 9 ijerph-20-00423-f009:**
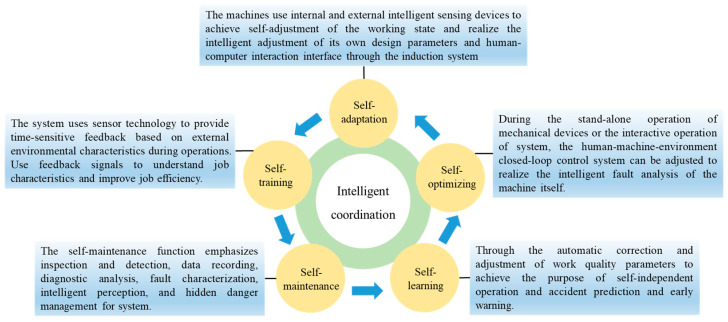
The function of intelligent coordination and its specific interpretation. By improving the intelligent coordination function of the system and integrating it into all aspects of system analysis, intrinsic safety and cleaner production can be realized.

**Figure 10 ijerph-20-00423-f010:**
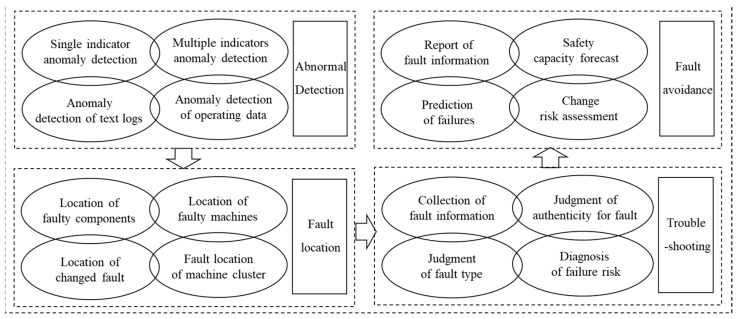
Dynamic decision-making component of intelligent equipment group operation and maintenance of the system. Dynamic decision-making components are based on the well-mined knowledge map of safe man-machine system operation and maintenance. It is an important means for realizing the intelligent control, operation, and maintenance of the equipment group.

## Data Availability

Not applicable.

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
