# Peer review of "Intelligent Safety Ergonomics: A Cleaner Research Direction for Ergonomics in the Era of Big Data"

_ijerph, 2022, doi:10.3390/ijerph20010423_

Round 1

Reviewer 1 Report

 As humans enter the era of big data, the development of information technology has brought new opportunities and challenges to the innovation, transformation, and upgrading of safety ergonomics.

It is a carefully done study and the findings are of considerable interest. I recommend the manuscript can be accepted. Detailed comments are listed below:

1. Compared with other safety disciplines, what are the outstanding characteristics of intelligent safety ergonomics?

 2. How to ensure the implementation of intrinsic safety? What are the theoretical implications of intelligent safety ergonomics? Please make it clearly.

3. Intelligent safety ergonomics has 5 basic functions to maintain the safe and stable operation and safety control of the system. Are they related? Please elaborate appropriately.

 4.What is the visual noise in human-computer interaction visualization? Do they lead people to behave in the wrong way in the process of working safety?

Author Response

Point 1

Compared with other safety disciplines, what are the outstanding characteristics of intelligent safety ergonomics?

Response 1

First of all, intelligent safety ergonomics extends the humans, machines, and environment concerned by the safety discipline to "humans, machines, environment and information", which is particularly important in the era of big data. Secondly, compared with traditional security disciplines, the tools used by intelligent safety ergonomics are more advanced and its theoretical basis is more cutting-edge. It takes cloud computing, big data, 5G, Internet of Things and intelligent sensing as tools and methods. Third, intelligent safety ergonomics maximizes the safety concept thoroughly into the design, implementation, operation, maintenance of the life cycle for system system. Fourthly, due to the forward-looking theory and tools, it is possible to achieve intrinsic safety.

Point 2

How to ensure the implementation of intrinsic safety? What are the theoretical implications of intelligent safety ergonomics? Please make it clearly  

Response 2

With the rapid development of information technology, intelligence gives us a special technical means, because the intelligent safety ergonomics adhering to integrate safety concept into the design, implementation, operation and maintenance of the human-machine loop system the whole life cycle, so as to achieve intrinsic safety.

By using the tools and theories of intelligent safety ergonomics, we can complete the safety measures that cannot be guaranteed by traditional technical means, and thus achieve intrinsic safety.

Modified content

Lines 207-211, where marked red in the file called “Revised Manuscript”.

Point 3

Intelligent safety ergonomics has 5 basic functions to maintain the safe and stable operation and safety control of the system. Are they related? Please elaborate appropriately.

Response 3

As the focus of intelligent safety ergonomics is to thoroughly integrate the safety concept into the full life cycle of the design, implementation, operation and maintenance of the humans, machines and environment, the five basic functions to maintain the safe and stable operation and safety control of the system, which complement each other and correspond to the full life cycle of the system, it is also an important means to achieve intrinsic safety.

Point 4

What is the visual noise in human-computer interaction visualization? Do they lead people to behave in the wrong way in the process of working safety?

Response 4

The visual noise in human-computer interaction visualization usually refers to the external conditions that hinder and mislead people to make the correct safe behaviours, and these factors usually lead to unsafe behaviours.

Reviewer 2 Report

The author's of manuscript proposes the basic concept of intelligent safety ergonomic (ISE), and analyzes  the characteristics, attributes, contents and research objects of ISE through theoretical analysis. The future challenges and unsolved problems of ISE are analyzed too.

The manuscript is clear, relevant for the field, a gap in knowledge is identified - the research studies on construction of ISE are not enough. I agree with author's conclusion "this study is a preliminary exploratory study of ISE and does not provide comprehensive answers to its theoretical questions, there is still a lack of suitable case studies on how production systems can be used effectively." I suggest for authors to continue studies on ISE by creating practically oriented algorithms of ISE application.

Some typing mistakes are noticed - lines 273, 370, 398, etc.

Author Response

Point 1

The manuscript is clear, relevant for the field, a gap in knowledge is identified - the research studies on construction of ISE are not enough. I agree with author's conclusion "this study is a preliminary exploratory study of ISE and does not provide comprehensive answers to its theoretical questions, there is still a lack of suitable case studies on how production systems can be used effectively." I suggest for authors to continue studies on ISE by creating practically oriented algorithms of ISE application.

Response 1

Thank you very much for your advice. Intelligent safety ergonomics, as a new discipline born of the cross synthesis of intelligent technology and traditional safety ergonomics, is an inevitable trend of subject iteration and update under the background in the era of big data, an inevitable product of the new round for subject revolution triggered by the development of artificial intelligence, and an inevitable trend of subject development in the era of big data. We are committed to applying the discipline theory into practice step by step and have achieved appropriate results. For example, combining with the function of the discipline, we use advanced monitoring devices to sense the risk factors in the mine production process, and ensure the safety of personnel and equipment through timely warning. Of course, there is still much work to be done in the development of the discipline.

In this revised manuscript, we also mentioned the related cases of ISE that guide practice, such as: identifying risk, risk positioning, intelligent monitoring and early warning, and adding relevant references to the manuscript. For example: in the life cycle of mine production, the degree of risk around it can be judged by monitoring the subtle changes in rocks, and then early implementation of safety control measures. It can be seen that use the tools and theories of intelligent safety ergo-nomics, we can complete the safety measures that cannot be guaranteed by traditional technical means, and thus achieve intrinsic safety (Lines 208-212). Other revised locations have been marked in red.

Modified content

Lines 207-211, where marked red in the file called “Revised Manuscript”.

Lines 208-212, Lines 390-392, Lines 403-406, Lines 496-499.

Point 2

Some typing mistakes are noticed - lines 273, 370, 398, etc.

Response 2

According to your suggestion, we have corrected the corresponding part. Thank you very much for your valuable advice in order to improve the quality of the manuscript.

Modified content

Lines 276-278, lines 374, and lines 401-404, where marked red in the file called “Revised Manuscript”.
